# The effect of results-based motivating system on metabolic risk factors of non-communicable diseases: A field trial study

**Mehran Asadi-Aliabadi**[1,2], **Seyed M. Karimi**[3], **Fariba Mirbaha-Hashemi**[2], **Arash Tehrani-Banihashemi**[2,4], **Leila Janani**[2], **Ebrahim Babaee**[2], **Marzieh Nojomi**[2,4], **Maziar Moradi-Lakeh**[5]*

**1** Health Sciences Research Center, Addiction Institute, Mazandaran University of Medical Sciences, Sari, Iran, **2** Preventive Medicine and Public Health Research Center, Psychosocial Health Research Institute, Iran University of Medical Sciences, Tehran, Iran, **3** Department of Health Management & System Sciences, School of Public Health & Information Sciences, University of Louisville, Louisville, Kentucky, United States of America, **4** Department of Community and Family Medicine, School of Medicine, Iran University of Medical Sciences, Tehran, Iran, **5** Gastrointestinal and Liver Diseases Research Center, Iran University of Medical Sciences, Tehran, Iran

* moradilakeh.m@iums.ac.ir, mazmoradi@gmail.com

## Abstract

### Background

Non-communicable diseases can be controlled and managed by reducing their associated metabolic risk factors. In this study, a set of intervention packages were designed to reduce the prevalence of three common metabolic risk factors (hypertension, hyperlipidemia, and obesity and overweight) in the community by motivating non-physician health workers.

### Methods

A field trial study was conducted in 4 districts of Iran. Thirty-two community health centers were randomly selected. A survey of 30 to 70-year-old was conducted to measure baseline metabolic risk factors. The intervention packages focused on improving hypertension, hyperlipidemia, obesity and overweight. The interventions included goal-setting, evidence-based education, operational planning, and incentive payments for non-physician health workers. A second survey to measure the final metabolic risk factors was performed after one year. The difference-in-difference method was used to evaluate the effectiveness of the intervention packages.

### Results

The average age of participants in both surveys was 49 years. The interventions had statistically significant effects only on decreasing the prevalence of overweight and obesity. The package with all the interventions except pay-for-performance decreased the odds of overweight and obesity to 0.57 (95% CI: 0.34, 0.95).

**Data Availability Statement:** The data supporting this study's findings cannot be shared publicly because of ethical and legal restrictions. Data are

available upon request from the Preventive Medicine and Public Health Research Center, Iran University of Medical Sciences or National Institute for Medical Research Development after appropriate protocol submission to the institution's office of Human Research Ethics Committee (contact via research.ethics.iums@gmail.com or research@ethics.ac.ir) for researchers who meet the criteria for access to confidential data. Prof. Maziar Moradi-Lakeh is head of the team and is responsible for considering data requests.

**Funding:** Research reported in this publication was supported by Elite Researcher Grant Committee under award number 958058 from the National Institutes for Medical Research Development (NIMAD), Tehran, Iran. MML received the award. The funders had no role in study design, data collection and analysis, decision to publish, or preparation of the manuscript.

**Competing interests:** The authors have declared that no competing interests exist.

## Conclusions

Involving non-physician health workers and having action plans based on the health needs of the covered population can decrease obesity and overweight in the community. However, longer trials are needed to observe the effects on hypertension and hyperlipidemia.

## Introduction

Non-communicable diseases (NCDs) cause approximately 80% of deaths worldwide and 83% in Iran [1, 2]. metabolic risk factors—e.g., hypertension, obesity and overweight (OB/OW), hypertriglyceridemia, high levels of low-density lipoprotein (LDL-C), low levels of high-density lipoprotein cholesterol (HDL-C), and glucose intolerance [3]—are associated with a range of NCDs, especially cardiovascular diseases and cancers. Globally, 10.4 million deaths and 218 million disability-adjusted life years (DALYs) were attributable to metabolic risk factors in 2017 [4]. According to the Global Burden of Disease 2019 report, 30.6% of all deaths in Iran were attributable to hypertension, 20.1% to high fasting plasma glucose, 18.8% to OB/OW, and 16.1% to hyperlipidemia. Also, 13.8% of all DALYs were attributable to hypertension, 12.9% to OB/OW, 11.5% to high fasting plasma glucose, and 7.8% to hyperlipidemia [5].

Given their remarkable public health risks, managing NCDs is essential. Considering the health care sector's limited budget in developing countries, cost-effective interventions are needed to achieve the global goal of 25% relative risk reduction in premature deaths from NCDs by 2025 [6] and the Sustainable Development Goals (SDG) target of reducing premature death from NCDs by one-third by 2030 [7].

The burden on NCDs can be reduced if their metabolic risk factors are managed. The leading modifiable risk factor for cardiovascular disease is hypertension [8], which can be asymptomatic. Approximately 20% [9] and sometimes up to 50% [10] of cases are not known by patients. A systematic review showed that every 10 mmHg reduction in blood pressure results in about 28% reduction in heart failure, 27% in stroke, 17% in coronary heart disease, and 13% in all-cause mortality rate [11]. Also, patients with hyperlipidemia often do not show any symptoms until atherosclerosis eventually leads to a heart attack or stroke [12], while approximately 51% of the burden of heart disease and 12% of the burden of ischemic stroke are attributable to high cholesterol [13]. OB/OW can increase the risk of various types of cancer [14], coronary heart disease, type 2 diabetes, gallstones, and disability [15, 16]. It is estimated that a one-kilogram weight loss can reduce the risk of diabetes by about 16% [17]; also, a 5% to 10% weight loss can reduce systolic and diastolic blood pressure levels by 5 mmHg [18], decrease triglyceride by 25 to 40 mg/dL [19], and increase HDL by 5 mg/dL [20]. Hence, investing in NCDs management by improving identification, screening, and treatment of NCDs can significantly improve community public health status [21]. Essential interventions can be provided through primary health care (PHC) providers to enhance timely diagnosis and treatment. Various intervention methods have been utilized which focus on PHC providers to control NCDs [22, 23]. Nonetheless, a 2018 World Health Organization (WHO) report indicated that more than 50% of countries are likely to miss their 2030 NCDs targets [24, 25].

In line with Iran's Health Transformation Program, the country's Ministry of Health and Medical Education (MOHME) established a Non-Communicable Diseases Committee (NCDC) in 2015 to develop evidence-based policies and monitor their proper implementation to achieve the related SDG targets [26]. In 2016, the NCDC developed the IraPEN program, adapted from the WHO Package of Essential Noncommunicable Disease Interventions (WHO

PEN) for PHC setting in low-resource areas [27]. Despite the nationwide scaling up of the PHC system under IraPEN, there have been challenges in the quality of implementation, stemming from insufficient knowledge and skills of care providers for building trust with clients (assurance dimension), failure to provide timely services (Responsiveness), and lack of empathy with clients [28].

This study adopted a set of strategies to overcome the aforementioned implementation challenges. The first strategy was goal-setting by holding workshops to inform non-physician health workers (NPHWs) about the status of NCDs' leading metabolic risk factors in their covered populations and the country's goal to reduce the risks. The second strategy was evidence-based training of NPHWs by introducing innovative interventions and cost-effective implementation methods. The third strategy was setting an action plan for NPHWs and providing ongoing advice during the implementation phase, along with financial support for implementation. The fourth strategy was incentivizing NPHWs based on their performance in meeting a set of pre-determined goals. This study aimed to measure the most effective strategies for preventing and controlling NCD metabolic risk factors (MetRFs).

## Materials and methods

### The design of the field trial

This study was part of a larger project and has been approved by the National Committee on Ethics in Medical Research (code: IR.NIMAD.REC.1396.084) as well as our institutional review board (code: IR.IUMS.REC.1395.1057613). This study's protocol has already been published (S1 File) [29] and was registered in the Iranian Registry of Clinical Trials (IRCT20081205001488N2) on 03/06/2018 (https://en.irct.ir/trial/774) (S2–S4 Files). This study was conducted according to CONsolidated Standards of Reporting Trials (CONSORT) guidelines (S1 Checklist). The ethical code of this study was issued based on the original protocol, but the post-changes were reviewed and approved by them (National Institute for Medical Research Development) (**S5 File**).

Four districts in Iran were randomly selected for the field trial: three districts as intervention districts and one as a non-intervention (control) district. The eligible districts for this study were those with at least four urban and four rural CHCs. The non-intervention district was Garmsar, 114 kilometers (km) southeast of Tehran (the country's capital city), with a covered population of 77,421, according to the country's 2016 census [30]. The intervention districts were Shahriar (46 km southwest of Tehran), Damghan (337 km northeast of Tehran), and Dashtestan (986 km south of Tehran), with a total 2016 population of 1,090,447 (Fig 1) [30].

Four urban and four rural community health centers (CHCs) were randomly selected in each of four selected districts, for a total of 32 centers. CHC inclusion criteria were described in detail in the published protocol [29]. The key inclusion criteria for a CHC were having at least two NPHWs who expressed willingness to participate in the study and to keep working at the CHC for at least two years. Physicians were not included in this trial because they were subjected to a nationwide performance-based payment program during the study [31].

In all selected CHCs, baseline surveys of the 30- to 70-year-old individuals were conducted from June to September 2018 to determine the existing status of MetRFs. During the surveys' data collection phase, only one male and one female household member from each of the following age group strata were interviewed: 30–39, 40–49, 50–59, and 60–70 years. For example, if two males between the ages of 30 and 39 years were living in a household, one of them was randomly selected and interviewed. Providing informed consent was a necessary inclusion criterion. The surveys were structured based on the Persian version of the WHO stepwise

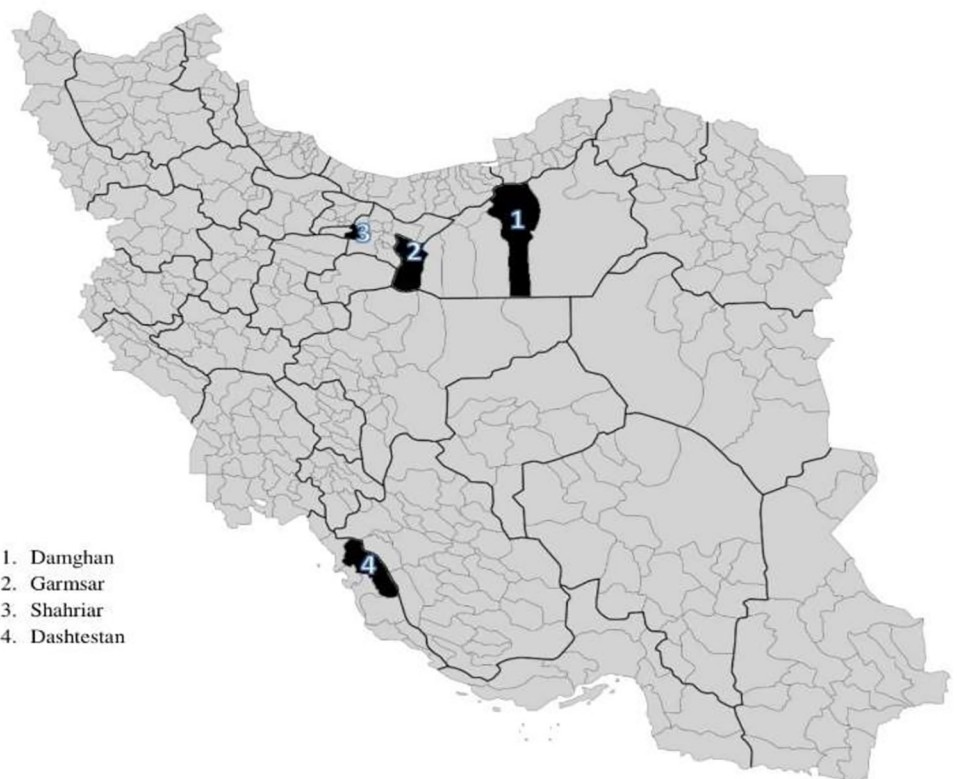

1. Damghan
2. Garmsar
3. Shahriar
4. Dashtestan

**Fig 1. Districts of intervention and no intervention in IRPONT study.**

approach to surveillance Questionnaire (STEPS) [32]. Subsequently, four different intervention packages were randomly assigned to the eight CHCs in each intervention district. One pair of urban and rural CHCs in each district received the first intervention package, and another pair received the second intervention package. After a 12-month period of intervention implementation, the second survey of NCDs MetRFs was conducted in all CHCs from September to November 2019 to evaluate the effectiveness of the intervention packages (Fig 2).

This study was planned for two years, and the third survey after 24 months of intervention was expected to be performed. However, it was terminated prematurely after 12 months of intervention due to the limitations imposed by the COVID-19 pandemic.

## Intervention packages

The basic intervention package (Package A) included only goal-setting seminars for NPHWs. During the workshops, after informing the participating NPHWs of the status of hypertension, hyperlipidemia, and OB/OW of the covered population based on the first survey, short-term targets such as specific reduction levels in the prevalence of the mentioned risk factors in the covered population and the country's national document to control and prevent NCDs and related risk factors were placed [33]. The targets were set based on the initial results of the baseline survey by researchers, local health authorities, and health staff.

The second intervention package (Package B) had an extra element in addition to goal-setting for NPHWs: it included evidence-based education of NPHWs through a two-day training workshop. The research team held the workshops at each of the districts. During the workshops, the effective and efficient interventions for preventing and controlling NCDs MetRFs

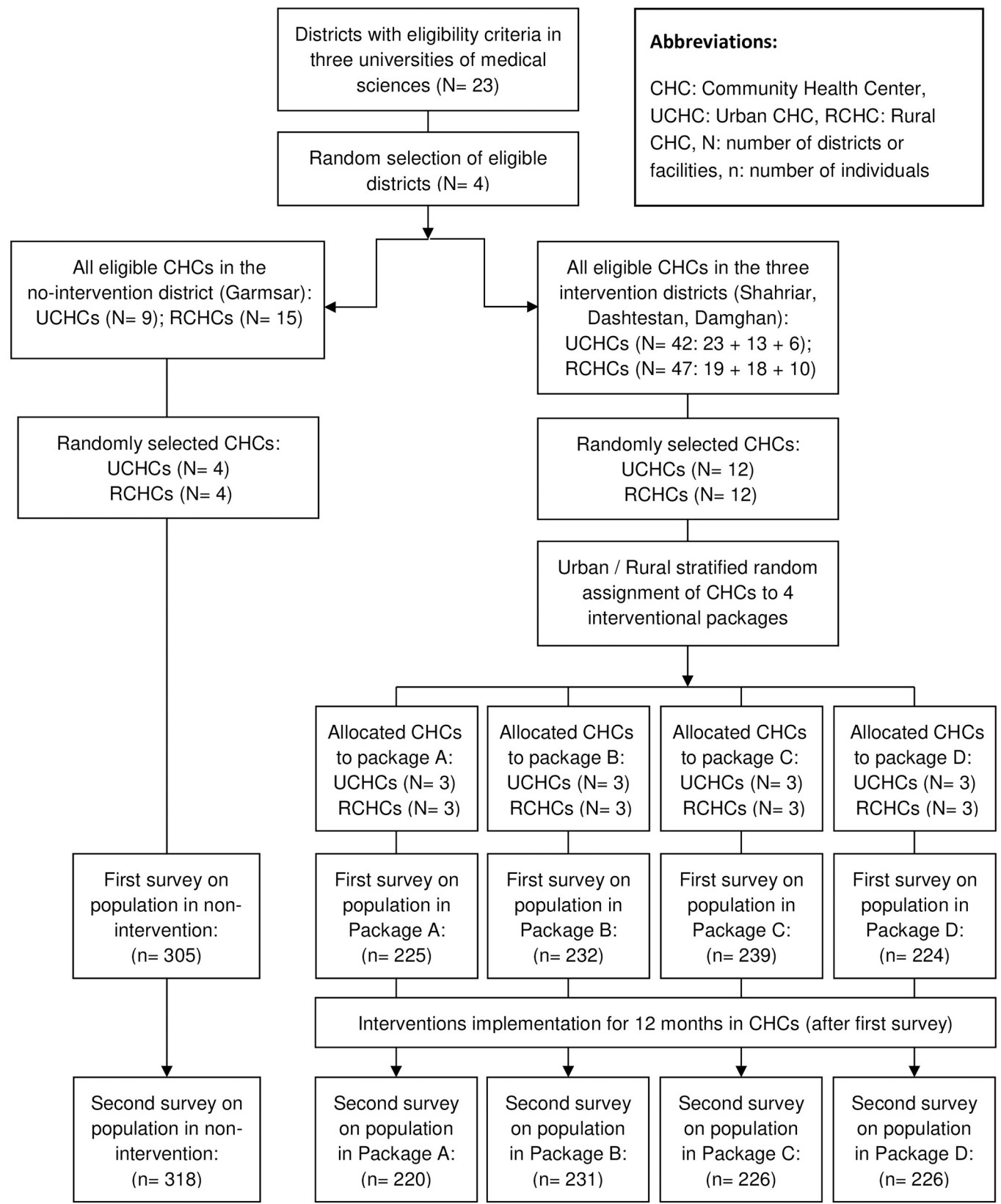

**Fig 2. CONSORT flow diagram.**

were presented and discussed, and the related documents were shared with NPHWs. The workshop training materials came from the World Bank's Disease Control Priorities, Volume 5, and the WHO PEN [22, 23]. The NPHWs made any decision on the selection, adoption, and execution of these interventions. While implementing intervention programs, the research team provided demand-based counseling services for NPHWs.

In the third intervention package (Package C), operational planning for NPHWs was added to goal-setting and evidence-based education. The research team and the NPHWs prepared an operational plan for each CHC based on the findings of the baseline survey for the CHC's covered population. For example, if OB/OW was the main MetRFs in the covered population, the plan emphasized actions that addressed OB/OW, like encouraging regular daily walking sessions in the community with the participation of NPHWs. The team also allocated budgets for the designed operational plans and consulted NPHWs during the plans' implementation. Also, the research team monitored intervention activities.

The fourth intervention package (Package D) added performance-based financing to the previous three elements. NPHWs received performance-based payments every three months without delay if they met the pre-determined goals. This data was collected from the electronic health system. If they reached at least 62.50% of their goals, they received the maximum incentive payment, 10% of a typical NPHW's average monthly salary. An NPHW's average monthly salary in 2018 was approximately 25 million Rials or 232 USD at a 107,832 Rial/USD current exchange rate [34]. If they reached 50.00% to 62.49% of their goals, NPHWs received an incentive equivalent to 8% of a typical NPHW's average monthly salary, 6% if they achieved from 25.00% to 49.99% of their goals. There was no incentive payment if NPHWs reached less than 25.00% of their goals. The incentive amounts were determined after consulting with districts and provincial supervisors and a focus group's discussions with NPHWs. The assignment of the intervention packages is shown in Fig 3.

Every two to four weeks, the implementation of interventions in the CHCs was monitored by the district and provincial supervisors. Similarly, the research team monitored the interventions every three months and collected related reports and documents.

## Statistical analysis

The objective of this study was to identify effective intervention packages by comparing the NCDs MetRFs before and after interventions within and between CHCs. NCDs MetRFs analyzed in this study were zero-one indicators of uncontrolled hypertension, hyperlipidemia, and OB/OW. Supplementary analyses were performed to compare the mean systolic and diastolic blood pressure, total cholesterol, and BMI before and after the interventions.

Criteria for identifying hypertension were systolic blood pressure (SBP) of 140 mmHg or higher, diastolic blood pressure (DBP) of 90 mmHg or higher, a physician or a nurse's diagnosis, or taking medication to control hypertension over the last fourteen days [35]. Criteria for identifying Hypercholesterolemia included a total cholesterol level of 200 mg/dL or higher, a physician or a nurse's diagnosis, or taking medication to control high cholesterol over the last fourteen days [36]. Body mass index (BMI) between 25.00 kg/m$^2$ and 29.99 kg/m$^2$ was defined as overweight and obese if greater than or equal to 30.00 kg/m$^2$ [37].

Prevalence and mean differences between the two surveys were calculated. Then, the difference-in-difference (DID) framework was used to identify the effect of the intervention packages. The following equation shows the linear specification of the DID design:

$$Y_{ict} = \alpha + \sum_{p=A,B,C,D} \beta^p \, Package^p_{ic} + \gamma Post_{it} + \sum_{p=A,B,C,D} \rho^p (Package^p_{ic} \times Post_{it}) + \sum_c \theta^c CHC^c_{it}$$
$$+ \delta X_{ict} + \varepsilon_{ict} \tag{1}$$

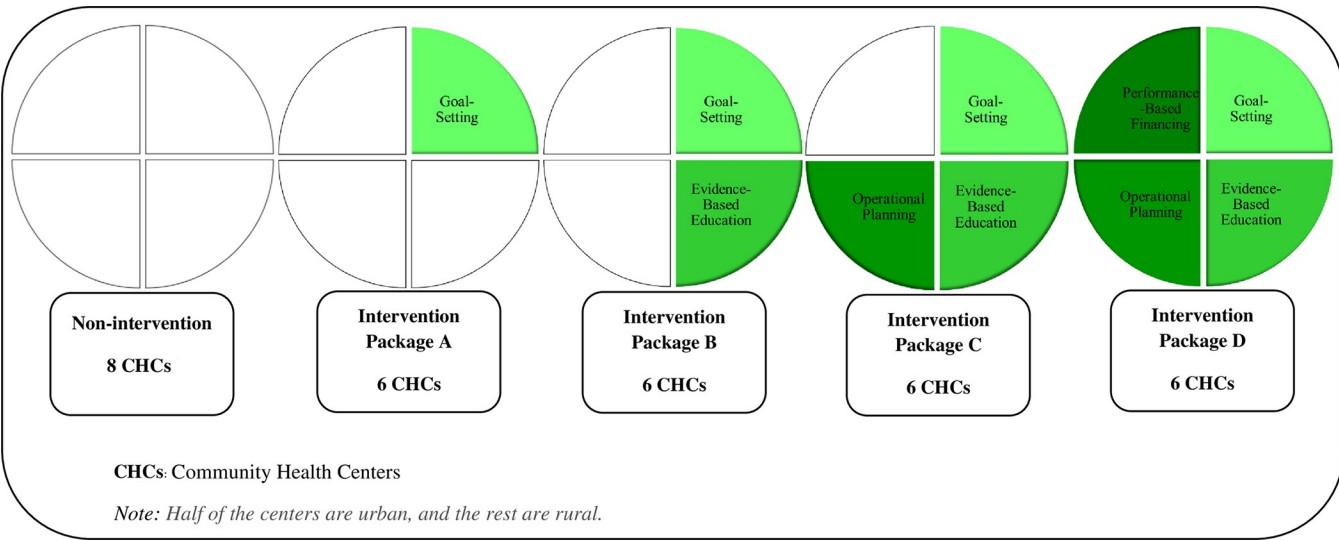

**Fig 3. Assignment of interventions to community health centers (CHCs) in districts.**

where $i$ indicates a surveyed individual, $c$ indicates the CHC through which the individual received PHC services, $t$ indicates the survey year, and $p$ indicates an intervention package. The dependent variable, $Y$, was a binary variable indicating one of the NCDs MetRFs or a continuous measure of NCDs MetRFs. The variable $Package^p$ (where $p$ takes characters A, B, C, and D, representing the four intervention packages) is a dummy variable that takes the value 1 if individual $i$, living in the catchment area of the community health center $c$, was exposed to the intervention package $p$, 0 otherwise. Although an ordered structure was considered when designing the intervention packages, we included the intervention package indicators as dummy variables, as the effects of the interventions might not necessarily be ordered. The variable $Post$ was set equal to 0 if individual $i$ participated in the baseline survey in 2018, 1 if participated in the post-intervention survey in 2019. The variable $CHC^c$ is a dummy variable that indicates living in the catchment area of the community health center $c$. Since 32 CHCs participated in this trial, we generated 32 dummy variables and included 31 of them in the statistical model to avoid multicollinearity. The variable $X$ is a vector of demographic and socioeconomic factors, including age, sex, marital status, education level, job status, health insurance status, and homeownership status. The estimated coefficients for the interaction of the variable $Package^p$ ($p$ = A, B, C, or D) and the variable $Post$ represent the effect of interest. That is, $\rho^A$, $\rho^B$, $\rho^C$, and $\rho^D$ measure the effect on $Y$ of the intervention packages A, B, C, and D, respectively, compared to the no intervention.

Given the binary and continuous nature of the outcome variables in this study, logistic and linear models were used in fitting Eq (1). Odds ratios and coefficients were calculated, representing the change in the dependent variable in association with changes in the explanatory variables of the equation. Standard errors were clustered at the CHC level to account for the possibility that the NCDs MetRFs may not be independently distributed within the population covered by each CHC [38]. Sampling weights were used in the regressions. The weights were the multiplication of two ratios. The first ratio was the share of sex-specific age groups (namely, 30–39, 40–49, 50–59, 60–70 years) in urban and rural areas in the country. The second ratio was the share of surveyed people of the same sex and age group in the individual's corresponding CHC, separately for urban and rural CHCs. The analysis was conducted with and without adjusting for the set of socioeconomic factors, $X$, to assess omitted variable bias

due to observables. The statistical package used for analyses was STATA 14.0 (STATA, Inc., College Station, Texas).

## Results

A total of 2446 individuals, 30- to 70-year-old, participated in the two surveys: 1225 in the first and 1221 in the second survey. The mean age of participants was 49.27 (SD of 0.33) and 49.38 (SD of 0.32) years in the first and second surveys, respectively. An almost equal number of men and women participated in the surveys. Women comprised 50.12% of the first survey and 50.37% of the second survey. The majority of the participants were illiterate or had primary school education: 43.48% and 42.45% in the first and second surveys, respectively. On the other hand, college-educated participants were in the minority: 10.76% of the first survey and 11.25% of the second survey. Most of the participants were married (85.42% and 85.44 in the first and second surveys), homeowners (83.85% in the first, 85.09% in the second survey), and homemakers (42.29% and 46.49% in the first and second surveys). Also, most of them had health insurance (92.58% in the first and 94.74% in the second survey) (**Table 1**).

The characteristics of participants across the study groups were largely similar. The most noticeable difference among the study groups was the low share of illiterate participants in the non-intervention group (25.83% in the first and 23.66% in the second survey) compared to the intervention groups (36.73% and 55.45%). Also, the percentage of homeowner participants in the intervention group receiving Package D (70.68% and 77.43% in the two surveys) was smaller than in other groups (79.17% and 92.89%). In addition, the self-employment rate across the groups fluctuates between 14.48% and 40.57%. Such differences highlight the importance of adjusting for socioeconomic characteristics in statistical analyses.

The crude difference in the prevalence of the studied NCDs MetRFs before and after the interventions was usually negative but rarely statistically significant (**Table 2**). The only statistically significant change was in hyperlipidemia. It occurred in CHCs that received the intervention package B (goal-setting and evidence-based training). Its prevalence decreased from 47.22% to 37.23%, a 9.99% decrease with a 95% confidence interval (CI) of 0.87% to 19.11% decrease. This was the largest decrease in the prevalence of the NCDs MetRFs. The second-largest decrease belonged to OB/OW and occurred in CHCs that received the intervention package C (goal-setting, evidence-based training, and operational planning). In these CHCs, the prevalence of OB/OW decreased from 72.41% to 66.22%, a 6.19% decrease (95% CI: -15.26%, 2.88%).

The before-after differences could have been influenced by the effect of time. If the effect of time on the studied outcomes was the same in the intervention and non-intervention CHCs, the DID design could identify the effects of the interventions independent of time. DID results, adjusted and unadjusted for socioeconomic characteristics, are presented in **Table 3**. The DID estimates generally showed a decrease in the odds of incidence of hypertension, hyperlipidemia, and OB/OW in the intervention CHCs versus the non-intervention CHCs. The odds of decrease, however, were statistically significant in only one case: the prevalence of OB/OW in CHCs that received the intervention package C (goal-setting, evidence-based training, and operational planning). Specifically, the adjusted DID estimates showed that the odds of reporting OB/OW in CHCs receiving package C versus non-intervention CHCs decreased to 0.57 (95% CI: 0.34, 0.95). Interestingly, one of the most significant decreases in the NCDs MetRFs prevalence rates also occurred for the same package and risk factor (**Table 2**).

Similar analyses with non-dichotomous outcome variables for the studied NCDs MetRFs, presented in the S6 File, showed no statistically significant effects of the interventions. Close-to-significant effect of intervention package C was measured only for systolic blood pressure and BMI, confirming the results for dichotomous outcome variables.

**Table 1. Demographic and economic characteristics of participants (frequencies and percentages).**

| Socioeconomic Factors | Total | | Intervention Package/Group | | | | | | | | | |
|---|---|---|---|---|---|---|---|---|---|---|---|---|
| | | | A | | B | | C | | D | | None | |
| | n = 2446 | | n = 445 | | n = 463 | | n = 465 | | n = 450 | | n = 623 | |
| | Survey 1 | Survey 2 | Survey 1 | Survey 2 | Survey 1 | Survey 2 | Survey 1 | Survey 2 | Survey 1 | Survey 2 | Survey 1 | Survey 2 |
| | n = 1225 | n = 1221 | n = 225 | n = 220 | n = 232 | n = 231 | n = 239 | n = 226 | n = 224 | n = 226 | n = 305 | n = 318 |
| Mean Age (Standard Deviation) | 49.3 (0.33) | 49.4 (0.32) | 49.5 (0.76) | 49.6 (0.74) | 49.3 (0.73) | 49.3 (0.76) | 49.9 (0.73) | 49.1 (0.74) | 48.9 (0.78) | 49.1 (0.73) | 48.8 (0.65) | 49.7 (0.63) |
| **p-value** | 0.83 | 0.97 | | | | | | | | | | |
| Sex: | | | | | | | | | | | | |
| Male (%) | 611 (49.9) | 606 (49.6) | 112 (49.8) | 107 (48.6) | 117 (50.4) | 117 (50.6) | 113 (47.3) | 107 (47.4) | 110 (49.1) | 117 (51.8) | 159 (52.1) | 158 (49.7) |
| Female (%) | 614 (50.1) | 615 (50.4) | 113 (50.2) | 113 (51.4) | 115 (49.6) | 114 (49.4) | 126 (52.7) | 119 (52.6) | 114 (50.9) | 109 (48.2) | 146 (47.9) | 160 (50.3) |
| **p-value** | 0.85 | 0.90 | | | | | | | | | | |
| Education: | | | | | | | | | | | | |
| Illiterate or Primary School (%) | 497 (43.5) | 517 (42.5) | 105 (50.0) | 122 (55.5) | 109 (49.1) | 116 (50.4) | 117 (52.5) | 121 (53.8) | 88 (47.3) | 83 (36.7) | 78 (25.8) | 75 (23.7) |
| Secondary School (%) | 158 (13.8) | 196 (16.1) | 19 (9.0) | 35 (15.9) | 39 (17.6) | 39 (17.0) | 20 (9.0) | 30 (13.3) | 27 (14.5) | 44 (19.5) | 53 (17.6) | 48 (15.1) |
| High School (%) | 365 (31.9) | 368 (30.2) | 65 (31.0) | 43 (19.6) | 58 (26.1) | 64 (27.8) | 68 (30.5) | 52 (23.1) | 45 (24.2) | 70 (31.0) | 129 (42.7) | 139 (43.9) |
| Some College (%) | 123 (10.8) | 137 (11.2) | 21 (10.0) | 2 (9.0) | 16 (7.2) | 11 (4.8) | 18 (8.0) | 22 (9.8) | 26 (14.0) | 29 (12.8) | 42 (13.9) | 55 (17.3) |
| **p-value** | <0.001 | <0.001 | | | | | | | | | | |
| Marital Status: | | | | | | | | | | | | |
| Never Married (%) | 82 (7.0) | 84 (6.9) | 12 (5.7) | 7 (3.2) | 9 (4.0) | 13 (5.7) | 18 (8.1) | 12 (5.4) | 19 (9.0) | 22 (9.7) | 24 (7.9) | 30 (9.4) |
| Married (%) | 996 (85.4) | 1039 (85.4) | 187 (89.5) | 198 (90.4) | 190 (86.0) | 194 (84.7) | 187 (84.2) | 198 (88.4) | 176 (83.0) | 185 (81.9) | 256 (84.8) | 264 (83.0) |
| Divorced/Widowed (%) | 88 (7.6) | 93 (7.7) | 10 (4.8) | 14 (6.4) | 22 (10.0) | 22 (9.6) | 17 (7.7) | 14 (6.2) | 17 (8.0) | 19 (8.4) | 22 (7.3) | 24 (7.6) |
| **p-value** | 0.46 | 0.58 | | | | | | | | | | |
| Job: | | | | | | | | | | | | |
| Public Wage and Salary (%) | 101 (8.7) | 75 (6.2) | 13 (6.3) | 8 (3.6) | 18 (8.1) | 14 (6.1) | 16 (7.2) | 15 (6.8) | 20 (9.4) | 13 (5.8) | 34 (11.4) | 25 (7.9) |
| Private Wage and Salary (%) | 104 (9.0) | 92 (7.6) | 21 (10.1) | 31 (14.0) | 21 (9.5) | 15 (6.5) | 28 (12.7) | 21 (9.6) | 12 (5.7) | 18 (8.1) | 22 (7.4) | 7 (2.2) |
| Self-Employed (%) | 316 (27.2) | 292 (24.1) | 53 (25.5) | 46 (20.8) | 54 (24.3) | 56 (24.6) | 32 (14.5) | 49 (22.3) | 86 (40.6) | 58 (26.0) | 91 (30.5) | 83 (26.1) |
| Homemaker (%) | 491 (42.3) | 563 (46.5) | 89 (42.8) | 103 (47.1) | 98 (44.1) | 110 (47.8) | 110 (49.8) | 105 (47.7) | 73 (34.4) | 102 (45.7) | 121 (40.6) | 143 (45.0) |
| Retired (%) | 91 (7.8) | 132 (10.9) | 20 (9.6) | 24 (10.9) | 19 (8.6) | 24 (10.4) | 22 (10.0) | 19 (8.6) | 12 (5.7) | 25 (11.2) | 18 (6.0) | 40 (12.6) |
| Unemployed (%) | 58 (5.0) | 57 (4.7) | 12 (5.8) | 8 (3.6) | 12 (5.4) | 11 (4.8) | 13 (5.9) | 11 (5.0) | 9 (4.25) | 7 (3.1) | 12 (4.0) | 20 (6.2) |
| **p-value** | 0.03 | 0.31 | | | | | | | | | | |
| Health Insurance: | | | | | | | | | | | | |
| Insured (%) | 1035 (92.6) | 1152 (94.7) | 203 (96.2) | 209 (95.4) | 209 (94.6) | 205 (90.3) | 171 (96.1) | 210 (92.9) | 164 (78.8) | 215 (95.1) | 288 (96.0) | 313 (98.4) |
| Uninsured (%) | 83 (7.4) | 64 (5.3) | 8 (3.8) | 10 (4.6) | 12 (5.4) | 22 (9.7) | 7 (3.9) | 16 (7.1) | 44 (21.2) | 11 (4.9) | 12 (4.0) | 5 (1.6) |
| **p-value** | <0.001 | <0.001 | | | | | | | | | | |
| Homeownership: | | | | | | | | | | | | |
| Yes (%) | 898 (83.8) | 1027 (85.1) | 183 (87.6) | 171 (79.2) | 170 (86.3) | 209 (92.9) | 164 (92.7) | 192 (86.1) | 135 (70.7) | 175 (77.4) | 246 (82.8) | 280 (88.3) |
| No (%) | 173 (16.2) | 180 (14.9) | 26 (12.4) | 45 (20.8) | 27 (13.7) | 16 (7.1) | 13 (7.3) | 31 (13.9) | 56 (29.3) | 51 (22.6) | 51 (17.2) | 37 (11.7) |
| **p-value** | <0.001 | <0.001 | | | | | | | | | | |

**Table 2. The difference in NCDs metabolic risk factors' prevalence between the two surveys.**

| NCDs Risk Factors | Intervention Package | First Survey (%) | Second Survey (%) | Difference (%) (95% Confidence Interval) |
|---|---|---|---|---|
| Hypertension | A | 38.38 | 36.36 | -2.02 (-11.16, 7.11) |
| | B | 47.84 | 43.72 | -4.12 (-13.43, 5.19) |
| | C | 36.58 | 38.05 | 1.47 (-7.67, 10.61) |
| | D | 37.50 | 34.07 | -3.43 (-12.45, 5.59) |
| | None | 31.25 | 35.01 | 3.76 (-3.63, 11.16) |
| Hyperlipidemia | A | 22.91 | 31.02 | 8.11 (-0.46, 16.67) |
| | B | 47.22 | 37.23 | -9.99 (-19.11, -0.87) |
| | C | 33.17 | 38.05 | 4.88 (-4.12, 13.88) |
| | D | 30.39 | 29.77 | -0.62 (-9.31, 8.07) |
| | None | 30.87 | 26.18 | -4.69 (-11.82, 2.45) |
| Obesity & Overweight | A | 70.44 | 73.61 | 3.17 (-5.43, 11.77) |
| | B | 71.76 | 67.39 | -4.37 (-12.89, 4.16) |
| | C | 72.41 | 66.22 | -6.19 (-15.26, 2.88) |
| | D | 72.54 | 69.91 | -2.63 (-11.19, 5.92) |
| | None | 69.39 | 71.47 | 2.08 (-5.18, 9.35) |

## Discussion

### An overview of the results

One-third of Iranian adults suffer from at least one of the NCDs MetRFs analyzed in this study [39]. Hence, it is essential to manage these MetRFs. We examined the effects of four low-cost intervention packages on the NCDs MetRFs by conducting a field trial. We involved NPHWs in implementing the interventions in a randomly selected set of CHCs in four districts. The basic package included setting goals for NPHWs to control NCD MetRFs in their CHCs.

**Table 3. Estimated effects of the intervention packages on the incidence of NCDs metabolic risk factors.**

| NCDs Risk Factor | Intervention Package | Unadjusted | | Adjusted for Socioeconomic Factors | |
|---|---|---|---|---|---|
| | | Odds Ratio (95% CI) | p-value | Odds Ratio (95% CI) | p-value |
| **Hypertension** | A | 0.66 (0.41, 1.07) | 0.09 | 0.69 (0.39, 1.22) | 0.20 |
| | B | 0.66 (0.29, 1.49) | 0.32 | 0.72 (0.34, 1.52) | 0.39 |
| | C | 1.07 (0.57, 2.04) | 0.82 | 0.88 (0.41, 1.91) | 0.75 |
| | D | 0.76 (0.40, 1.44) | 0.40 | 0.64 (0.29, 1.41) | 0.27 |
| | None | Reference Group | | | |
| **Hyperlipidemia** | A | 1.50 (0.62, 3.65) | 0.37 | 1.78 (0.66, 4.48) | 0.26 |
| | B | 0.77 (0.31, 1.95) | 0.58 | 0.95 (0.34, 2.68) | 0.93 |
| | C | 1.71 (0.81, 3.59) | 0.16 | 1.65 (0.70, 3.89) | 0.25 |
| | D | 1.18 (0.50, 2.76) | 0.71 | 1.39 (0.45, 4.30) | 0.57 |
| | None | Reference Group | | | |
| **Obesity & Overweight** | A | 0.96 (0.67, 1.35) | 0.80 | 0.97 (0.67, 1.40) | 0.86 |
| | B | 0.76 (0.37, 1.56) | 0.45 | 0.68 (0.31, 1.50) | 0.34 |
| | C | 0.72 (0.45, 1.14) | 0.16 | 0.57 (0.34, 0.95) | 0.03 |
| | D | 0.83 (0.38, 1.79) | 0.63 | 0.72 (0.33, 1.55) | 0.40 |
| | None | Reference Group | | | |

Other packages had additive components of evidence-based training, operational planning, and performance-based financing for NPHWs. Improvements were observed in OB/OW, and the most effective intervention package was package C, which included all interventions but performance-based financing. Neither of the packages had a statistically significant improvement effect on the prevalence of hypertension. Also, no improvement in the effect of the intervention packages on hyperlipidemia and total cholesterol levels was observed.

## Assessing BMI-related results

This study's estimated improvements in BMI-related outcomes (namely, the incidence of OB/OW and the level of BMI) can be attributed to exercise and healthy diet training sessions organized by NPHWs in the studied communities. In communities without adequate facilities for physical activity, emphasis was placed on proper diet and counseling sessions. Previous research has shown that NPHWs' availability and proper communication with their clients are effective in behavioral changes in the population [40, 41]. Another important factor in the effectiveness of community-based behavioral interventions in reducing OB/OW is the duration of interventions. This analysis measured signs of effectiveness after one year of intervention. A review study of 6 articles reported a significant reduction in OB/OW after an average of 10 months of population-based lifestyle interventions [42]. In another review of 21 studies, significant OB/OW reductions in population-based lifestyle interventions were reported over 6 to 24 months [43]. In addition, a quasi-experimental study in Malaysia showed that community-based lifestyle interventions for 6 months had no effect on weight loss and obesity, but significant effects were observed after 12 months [44].

## Assessing blood pressure-related results

No significant effect of the intervention packages on uncontrolled hypertension was measured in this study. The limited effect of the interventions on the blood pressure level can be explained by NPHWs' lack of authority to prescribe blood pressure medicine drugs and not receiving adequate supervision from physicians. Community-based interventions in Colombia and Malaysia showed that behavioral and therapeutic interventions supervised by local physicians were effective in improving blood pressure within 12 months [45]. Also, 12-month lifestyle consultations by NPHWs supervised by physicians led to SBP reduction in China [46]. Studies in Tunisia and Pakistan showed that implementing healthy lifestyle interventions for 2 to 3 years via CHCs and effective use of local resources and networks decreased the SBP in the community [47, 48]. The difference between these studies and our study was physician supervision of interventions, authority in drug prescription, and more time to implement interventions.

## Assessing cholesterol-related results

No significant effect of the studied interventions was measured in reducing total cholesterol and the incidence of hyperlipidemia. This result may also pertain to the lack of medication interventions by NPHWs and no physician supervision. In contrast to this study, the effectiveness of behavioral lifestyle interventions was documented when accompanied by therapeutic interventions [49, 50] and supervised by physicians [51, 52]. Another reason for not observing an effect on cholesterol-related outcomes may be the short one-year period of the interventions. Longer-term interventions could effectively lower total cholesterol [53–58]. For example, the North Carolina Project [57] and Coalfields Healthy Heartbeat [58] showed effectiveness in lowering total cholesterol levels after 5 and 6 years, respectively.

## Features of successful community-based interventions

The success of community-based interventions in decreasing NCDs MetRFs largely depends on the duration of interventions [59–62]. Duration is important because changes in NCDs MetRFs usually follow a specific sequence (e.g., hypertension follows OB/OW) [63–68]. Another critical determining factor is the extent of cross-sectoral and intra-sectoral coordination [69, 70]. Using and coordinating various community resources (e.g., health care facilities other than CHCs, schools' networks, religious institutions, city and village councils, and non-governmental organizations) can advance messaging and incentivize public health improvements. Also, as NPHWs are not authorized to prescribe drugs, coordination with and supervision of physicians are essential when medication interventions are needed. In the Iranian health system, NPHWs are authorized to educate and encourage people for lifestyle modifications and follow up on physicians' prescriptions of medications [71]. Thus, their contributions to local health policies and the planning of health interventions are mainly indirect. However, the results of this and other studies showed that managing risk factors through NPHWs can result in significant improvements if combined with lifestyle and treatment interventions [72–74].

## Elements of community-based interventions

This research showed that a low-cost, pragmatic intervention that includes evidence-based training and innovative action plans could reduce OB/OW in the short term. Other studies have shown that holding training courses for health workers along with having operational plans with goals in the short, medium, and long term can have a significant impact on improving NCDs MetRFs [58, 75–78]. On the other hand, performance-based payment can create an infrastructure to improve performance. A potentially important factor in the ineffectiveness of incentive payments in this study was the insignificant amount of incentive payments: the maximum incentive payment was only 10% of a typical NPHW's average monthly salary, about 23 USD, every three months. Other studies have shown that better results were achieved with higher incentive payments [79–81]. Although the intervention packages were ordered from simple to complex, the observed effects did not necessarily change this way.

## Limitations

Several limitations potentially affected this study's results. First, this was not a longitudinal study, as the program's participants were not followed throughout its implementation. The first and second surveys were snapshots of the populations covered by the selected CHC. The survey populations included non-participating individuals as well as participating ones. Consequently, the measured effects of the intervention packages may underestimate their real effects. Second, the study was cut short because of the COVID-19 pandemic. Reducing metabolic risk factors usually takes persistence and will gradually materialize. A one-year period may not be sufficient to observe major changes in MetRFs. Therefore, similar studies conducted over multiple years need to be conducted to assess the effects of such interventions on MetRFs. Third, the researchers have no control over the relocation of NPHWs in the selected CHCs. The relocation could affect the consistency of service provision at the CHCs. Nonetheless, all CHCs where a relocation occurred were immediately identified, and similar training was provided to substituting NPHWs. Fourth, it is possible that concurrent national research projects affected this study by influencing the NPHWs' effort level and priorities. One such project was the country's High Blood Pressure Campaign [82].The influence of such national-level factors on non-intervention and different intervention groups, however, might be similar.

## Conclusion

Having action plans based on the health needs of the covered population and involving NPHWs in the plans, as well as holding retraining courses and introducing new strategies in the prevention and control of NCDs MetRFs are effective steps in controlling OB/OW in the community. This study was shortened because of the emergence of the COVID-19 pandemic. Improvements in other NCDs MetRFs might have been measured had the interventions been implemented for more than one year.

## Supporting information

**S1 Checklist. CONSORT checklist.**
(DOCX)

**S2 Checklist.** *PLOS ONE* **clinical studies checklist.**
(DOCX)

**S1 File. Study protocol.**
(PDF)

**S2 File. RCT_First_Register_protocol.**
(PDF)

**S3 File. RCT_Second_Register_protocol.**
(PDF)

**S4 File. RCT_Last_Register_protocol.**
(PDF)

**S5 File. Protocol deviations.**
(PDF)

**S6 File. Supplementary tables.**
(PDF)

**S7 File. IRPONT collaborators.**
(PDF)

## Acknowledgments

The authors would like to thank the vice-chancellors of Public Health at Iran University of Medical Sciences (Tehran, Iran), Semnan University of Medical Sciences (Semnan, Iran), and Bushehr University of Medical Sciences (Bushehr, Iran) and the involved health care workers of Shahriar, Damghan, Garmsar, and Borazjan for their contribution to this project. A full list of the collaborators of this study is available in the electronic S7 File.

## Author Contributions

**Conceptualization:** Maziar Moradi-Lakeh.

**Data curation:** Mehran Asadi-Aliabadi, Maziar Moradi-Lakeh.

**Formal analysis:** Mehran Asadi-Aliabadi, Seyed M. Karimi, Leila Janani.

**Investigation:** Mehran Asadi-Aliabadi, Arash Tehrani-Banihashemi, Ebrahim Babaee, Maziar Moradi-Lakeh.

**Methodology:** Mehran Asadi-Aliabadi, Arash Tehrani-Banihashemi, Marzieh Nojomi, Maziar Moradi-Lakeh.

**Supervision:** Fariba Mirbaha-Hashemi, Arash Tehrani-Banihashemi, Maziar Moradi-Lakeh.

**Writing – original draft:** Mehran Asadi-Aliabadi, Seyed M. Karimi, Maziar Moradi-Lakeh.

**Writing – review & editing:** Mehran Asadi-Aliabadi, Seyed M. Karimi, Fariba Mirbaha-Hashemi, Arash Tehrani-Banihashemi, Leila Janani, Ebrahim Babaee, Marzieh Nojomi, Maziar Moradi-Lakeh.

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
