## [Decision Letter · Decision Letter 0]

10 Nov 2023

PONE-D-23-20016The effect of results-based motivating system on metabolic risk factors of non-communicable diseases: a field trial studyPLOS ONE

Dear Dr. Moradi-Lakeh,

Thank you for submitting your manuscript to PLOS ONE. After careful consideration, we feel that it has merit but does not fully meet PLOS ONE’s publication criteria as it currently stands. Therefore, we invite you to submit a revised version of the manuscript that addresses the points raised during the review process.

Please note that we have only been able to secure a single reviewer to assess your manuscript. We are issuing a decision on your manuscript at this point to prevent further delays in the evaluation of your manuscript. Please be aware that the editor who handles your revised manuscript might find it necessary to invite additional reviewers to assess this work once the revised manuscript is submitted.  The reviewer raises a concern about the statistical model used. Could you please carefully revise the manuscript to address this issue?

We look forward to receiving your revised manuscript.

Kind regards,

Steve Zimmerman, PhD

Senior Editor, PLOS ONE

Journal Requirements:

"Research reported in this publication was supported by Elite Researcher Grant

Committee under award number 958058 from the National Institutes for Medical Research Development (NIMAD), Tehran, Iran. MML received the award."

4. We note that the original protocol that you have uploaded as a Supporting Information file contains an institutional logo. As this logo is likely copyrighted, we ask that you please remove it from this file and upload an updated version upon resubmission.

Reviewers' comments:

Reviewer's Responses to Questions

**Comments to the Author**

1. Is the manuscript technically sound, and do the data support the conclusions?

Reviewer #1: Partly

2. Has the statistical analysis been performed appropriately and rigorously? 

Reviewer #1: No

3. Have the authors made all data underlying the findings in their manuscript fully available?

Reviewer #1: Yes

4. Is the manuscript presented in an intelligible fashion and written in standard English?

Reviewer #1: Yes

5. Review Comments to the Author

Reviewer #1: Recommedation

Major Revision Required

I have no problems with the objectives or design of this well conducted trial. However, the authors should consider whether a simpler presentation of their results would suffice.

The authors describe what is clearly a well-designed and conducted field trial and should be congratulated for this. However, I am not convinced that the statistical model of line 202 is entirely appropriate. Thus, for example, Table 3 (line 310) shows the hypertension outcome in which the successive differences quoted for packages None, A, B, C and D are +3.76, −2.02, −4,12, 1.47, −3.43. These are suggestive that Package does not have a linear effect and should not be considered as an ordered categorical variable as is indicated by the β in the statistical model (lines 202 and 207) but rather treated with dummy variables. Also, in the model CHC is regarded as an ordered categorical variable (coefficient θ) whereas it too should be modelled with dummy variables. Although this may cause some problems as the CHCs are nested within their respective packages. However, I do not think the broad conclusions are likely to be seriously affected but rather suggest that summarising the results as in Table 3 for example, (rather than fitting regression models) may be sufficient for this paper.

6. PLOS authors have the option to publish the peer review history of their article (what does this mean?). If published, this will include your full peer review and any attached files.

Reviewer #1: No

---

## [Author Response · Author response to Decision Letter 0]

23 Jan 2024

PONE-D-23-20016

The effect of results-based motivating system on metabolic risk factors of non-communicable diseases: a field trial study

PLOS ONE

Dear Dr. Moradi-Lakeh,

Thank you for submitting your manuscript to PLOS ONE. After careful consideration, we feel that it has merit but does not fully meet PLOS ONE’s publication criteria as it currently stands. Therefore, we invite you to submit a revised version of the manuscript that addresses the points raised during the review process.

Please note that we have only been able to secure a single reviewer to assess your manuscript. We are issuing a decision on your manuscript at this point to prevent further delays in the evaluation of your manuscript. Please be aware that the editor who handles your revised manuscript might find it necessary to invite additional reviewers to assess this work once the revised manuscript is submitted. 

 The reviewer raises a concern about the statistical model used. Could you please carefully revise the manuscript to address this issue?

We look forward to receiving your revised manuscript.

Kind regards,

Steve Zimmerman, PhD

Senior Editor, PLOS ONE

Journal Requirements:

Response 1: We addressed all the PLOS ONE style requirements.

"Research reported in this publication was supported by Elite Researcher Grant Committee under award number 958058 from the National Institutes for Medical Research Development (NIMAD), Tehran, Iran. MML received the award."

Response 2: The funders had no role in study design, data collection and analysis, decision to publish, or preparation of the manuscript

Response 3: The data supporting this study’s findings cannot be shared publicly because of ethical and legal restrictions. Data are available upon request from the Preventive Medicine and Public Health Research Center, Iran University of Medical Sciences or National Institute for Medical Research Development after appropriate protocol submission to the institution’s office of Human Research Ethics Committee (contact via research.ethics.iums@gmail.com or research@ethics.ac.ir) for researchers who meet the criteria for access to confidential data. Prof. Maziar Moradi-Lakeh is head of the team and is responsible for considering data requests.

4. We note that the original protocol that you have uploaded as a Supporting Information file contains an institutional logo. As this logo is likely copyrighted, we ask that you please remove it from this file and upload an updated version upon resubmission.

Response 4: The Medical Journal of the Islamic Republic of Iran (MJIRI) is an editorially independent peer-reviewed online open-access journal owned and published by Iran University of Medical Sciences (IUMS), and this institutional logo belongs to IUMS.

Reviewers' comments:

Reviewer's Responses to Questions

Comments to the Author

1. Is the manuscript technically sound, and do the data support the conclusions?

Reviewer #1: Partly

2. Has the statistical analysis been performed appropriately and rigorously?

Reviewer #1: No

3. Have the authors made all data underlying the findings in their manuscript fully available?

Reviewer #1: Yes

4. Is the manuscript presented in an intelligible fashion and written in standard English?

Reviewer #1: Yes

5. Review Comments to the Author

Reviewer #1: Recommedation

Major Revision Required

I have no problems with the objectives or design of this well conducted trial. However, the authors should consider whether a simpler presentation of their results would suffice.

The authors describe what is clearly a well-designed and conducted field trial and should be congratulated for this. However, I am not convinced that the statistical model of line 202 is entirely appropriate. Thus, for example, Table 3 (line 310) shows the hypertension outcome in which the successive differences quoted for packages None, A, B, C, and D are +3.76, −2.02, −4,12, 1.47, −3.43. These are suggestive that Package does not have a linear effect and should not be considered as an ordered categorical variable as is indicated by the β in the statistical model (lines 202 and 207) but rather treated with dummy variables. Also, in the model CHC is regarded as an ordered categorical variable (coefficient θ) whereas it too should be modelled with dummy variables. Although this may cause some problems as the CHCs are nested within their respective packages. However, I do not think the broad conclusions are likely to be seriously affected but rather suggest that summarising the results as in Table 3 for example, (rather than fitting regression models) may be sufficient for this paper.

Response: Thank you very much for your careful review of the article. The statistical model presented in line 202 was used to estimate the effect of the four interventional packages on NCD Metabolic risk factors (NCDs MetRFs) applied to different sets of community health centers (CHC). This is a standard difference-in-difference specification in which the timing of potential exposure to the intervention is captured by the dummy variable named “Post”, and living in the catchment area of intervention CHCs is captured by a set of dummy variables represented by the vector “Package”. The effects of interest, then, are measured by the coefficients of the interaction of these two variables (Package × Post). 

Although it is not immediately clear from the specification of the statistical model, we generated dummy variables to represent these packages: 

 The first dummy variable took the value 1 for all individuals surveyed in the no intervention CHCs, 0 otherwise; 

 The second dummy variable took the value 1 for all individuals surveyed in CHCs that received the intervention package A, 0 otherwise; 

 The third dummy variable took the value 1 for all individuals surveyed in CHCs that received the intervention package B, 0 otherwise;

 The fourth dummy variable took the value 1 for all individuals surveyed in CHCs that received the intervention package C, 0 otherwise; 

 The fifth dummy variable took the value 1 for all individuals surveyed in CHCs that received the intervention package D, 0 otherwise; 

We did not include the first dummy variable in the statistical model to avoid multicollinearity. Therefore, we estimated four βs and four ρs. Since ρs represent the effects of interest (i.e., the association of exposure to a specific intervention package), we only reported them in Table 4. 

In addition, we included a separate dummy variable for each CHC to adjust for the effect of non-observable time-invarying factors specific to a CHC. 

In sum, we used dummy variables to indicate different intervention packages and CHCs in the statistical model, while it appears we included them linearly. We have now revised the statistical model to reflect the exact variable specifications. We have also provided more descriptions for the statistical model in the text.

6. PLOS authors have the option to publish the peer review history of their article (what does this mean?). If published, this will include your full peer review and any attached files.

Do you want your identity to be public for this peer review? For information about this choice, including consent withdrawal, please see our Privacy Policy.

Reviewer #1: No

Response: The figures revised by PACE.

---

## [Decision Letter · Decision Letter 1]

19 Mar 2024

PONE-D-23-20016R1The effect of results-based motivating system on metabolic risk factors of non-communicable diseases: a field trial studyPLOS ONE

Dear Dr. Moradi-Lakeh,

Thank you for submitting your manuscript to PLOS ONE. After careful consideration, we feel that it has merit but does not fully meet PLOS ONE’s publication criteria as it currently stands. Therefore, we invite you to submit a revised version of the manuscript that addresses the points raised during the review process.

We look forward to receiving your revised manuscript.

Kind regards,

Thaís São-João

Academic Editor

PLOS ONE

Additional Editor Comments:

Dear Maziar Moradi-Lakeh,

I would like to express our appreciation for your contribution to our journal and for the effort you have put into this work.

After careful consideration by our editorial board and external reviewers, I regret to inform you that your manuscript requires major revisions before it can be considered for publication. While your study addresses an important topic and presents potentially valuable insights, there are several significant issues that need to be addressed to enhance the quality and rigor of the paper.

Specifically, the reviewers have highlighted the following concerns:

1. Clarity and Organization: The overall organization of the manuscript needs improvement. The flow of ideas is not consistently logical, and certain sections lack clarity, making it challenging for readers to follow the argumentation.

2. Methodological Rigor: There are concerns regarding the methodology employed in the study. Certain aspects of the experimental design or data analysis need further clarification to ensure the validity and reliability of the findings.

3. Literature Review: The literature review provided in the manuscript could be more comprehensive. It is essential to provide a thorough review of relevant prior research to contextualize your study and demonstrate its novelty and significance.

4. Discussion and Conclusions: The discussion section requires further elaboration and interpretation of the results. Additionally, the conclusions drawn should be more firmly grounded in the findings of the study and supported by evidence.

I understand that revising the manuscript may require significant time and effort on your part, but I assure you that addressing these concerns will greatly enhance the overall quality and impact of your work. Therefore, I would like to invite you to revise your manuscript in accordance with the reviewers' feedback.

Thank you once again for considering PLOS ONE for the publication of your work. We look forward to receiving your revised manuscript.

Sincerely,

Reviewers' comments:

Reviewer's Responses to Questions

**Comments to the Author**

1. If the authors have adequately addressed your comments raised in a previous round of review and you feel that this manuscript is now acceptable for publication, you may indicate that here to bypass the “Comments to the Author” section, enter your conflict of interest statement in the “Confidential to Editor” section, and submit your "Accept" recommendation.

Reviewer #1: (No Response)

Reviewer #2: (No Response)

2. Is the manuscript technically sound, and do the data support the conclusions?

Reviewer #1: Yes

Reviewer #2: Partly

3. Has the statistical analysis been performed appropriately and rigorously? 

Reviewer #1: Yes

Reviewer #2: No

4. Have the authors made all data underlying the findings in their manuscript fully available?

Reviewer #1: Yes

Reviewer #2: Yes

5. Is the manuscript presented in an intelligible fashion and written in standard English?

Reviewer #1: Yes

Reviewer #2: Yes

6. Review Comments to the Author

Reviewer #1: Minor Revision

The authors have addressed my concern re the statistical model in the earlier version of this paper. The problem was that in the modelling ‘ordered categorical’ assumptions were made where the use of ‘dummy variables’ would have been more appropriate. However, if as seems to be the case here (see Page 1, line 35), when using a ‘dummy variable’ approach essentially gives the same conclusions as the ‘ordered categorical’ approach then to present the results with this (simpler) modelling approach seems sensible. In such circumstances, the authors should make it clear in their presentation what has been done. So, I suggest the authors revert back to their model as expressed in Line 202 of their original submission but add a sentence or two to explain that the ‘categorical’ approach leads to the same conclusions than the more complex ‘dummy' approach. I apologise to the authors for raising this technical point as it does not seem to affect their conclusions.

A minor comment to add. Reduce p-values to 2 significant figures in Table 2 (and elsewhere). For example, ‘0.831 to 0.83’ and ‘0.968 to 0.99’. However, for p-values of less than 0.001 the notation p <0.001 is acceptable. There appear to be some misprints in the table as a p-value given as ‘0.001>’ should be ‘<0.001’.

Reviewer #2: The study presents a well-defined and appropriately conducted research problem, objective and methods. However, the interventions carried out are not clearly presented (method section). The results contain a dense arrangement of data that is also unclear. I suggest that the presentation of the results is simple and summarized. There is a lot of information and it confuses reading.

I also believe that the statistical model presented is not the best. Why use a dummy model? The way the results were presented is very confusing.

The first reviewer made this consideration, however in a new round the authors did not adjust the manuscript in order to make it simpler.

Therefore, I recommend that the article be rejected and the authors prepare a new version for another magazine in order to provide clarity and lightness in the presentation of information.

7. PLOS authors have the option to publish the peer review history of their article (what does this mean?). If published, this will include your full peer review and any attached files.

Reviewer #1: No

Reviewer #2: No

---

## [Author Response · Author response to Decision Letter 1]

5 Apr 2024

Date: Mar 19 2024 09:31PM

To: "Maziar Moradi-Lakeh" moradilakeh.m@iums.ac.ir

From: "PLOS ONE" plosone@plos.org

Subject: PLOS ONE Decision: Revision required [PONE-D-23-20016R1]

PONE-D-23-20016R1

The effect of results-based motivating system on metabolic risk factors of non-communicable diseases: a field trial study

PLOS ONE

Dear Dr. Moradi-Lakeh,

Thank you for submitting your manuscript to PLOS ONE. After careful consideration, we feel that it has merit but does not fully meet PLOS ONE’s publication criteria as it currently stands. Therefore, we invite you to submit a revised version of the manuscript that addresses the points raised during the review process.

We look forward to receiving your revised manuscript.

Kind regards,

Thaís São-João

Academic Editor

PLOS ONE

Additional Editor Comments:

Dear Maziar Moradi-Lakeh,

I would like to express our appreciation for your contribution to our journal and for the effort you have put into this work.

After careful consideration by our editorial board and external reviewers, I regret to inform you that your manuscript requires major revisions before it can be considered for publication. While your study addresses an important topic and presents potentially valuable insights, there are several significant issues that need to be addressed to enhance the quality and rigor of the paper.

Specifically, the reviewers have highlighted the following concerns:

1. Clarity and Organization: The overall organization of the manuscript needs improvement. The flow of ideas is not consistently logical, and certain sections lack clarity, making it challenging for readers to follow the argumentation.

2. Methodological Rigor: There are concerns regarding the methodology employed in the study. Certain aspects of the experimental design or data analysis need further clarification to ensure the validity and reliability of the findings.

3. Literature Review: The literature review provided in the manuscript could be more comprehensive. It is essential to provide a thorough review of relevant prior research to contextualize your study and demonstrate its novelty and significance.

4. Discussion and Conclusions: The discussion section requires further elaboration and interpretation of the results. Additionally, the conclusions drawn should be more firmly grounded in the findings of the study and supported by evidence.

I understand that revising the manuscript may require significant time and effort on your part, but I assure you that addressing these concerns will greatly enhance the overall quality and impact of your work. Therefore, I would like to invite you to revise your manuscript in accordance with the reviewers' feedback.

Thank you once again for considering PLOS ONE for the publication of your work. We look forward to receiving your revised manuscript.

Sincerely,

Reviewers' comments:

Reviewer's Responses to Questions

Comments to the Author

1. If the authors have adequately addressed your comments raised in a previous round of review and you feel that this manuscript is now acceptable for publication, you may indicate that here to bypass the “Comments to the Author” section, enter your conflict of interest statement in the “Confidential to Editor” section, and submit your "Accept" recommendation.

Reviewer #1: (No Response)

Reviewer #2: (No Response)

2. Is the manuscript technically sound, and do the data support the conclusions?

Reviewer #1: Yes

Reviewer #2: Partly

3. Has the statistical analysis been performed appropriately and rigorously?

Reviewer #1: Yes

Reviewer #2: No

4. Have the authors made all data underlying the findings in their manuscript fully available?

Reviewer #1: Yes

Reviewer #2: Yes

5. Is the manuscript presented in an intelligible fashion and written in standard English?

Reviewer #1: Yes

Reviewer #2: Yes

6. Review Comments to the Author

Reviewer #1: Minor Revision

The authors have addressed my concern re the statistical model in the earlier version of this paper. The problem was that in the modelling ‘ordered categorical’ assumptions were made where the use of ‘dummy variables’ would have been more appropriate. However, if as seems to be the case here (see Page 1, line 35), when using a ‘dummy variable’ approach essentially gives the same conclusions as the ‘ordered categorical’ approach then to present the results with this (simpler) modelling approach seems sensible. In such circumstances, the authors should make it clear in their presentation what has been done. So, I suggest the authors revert back to their model as expressed in Line 202 of their original submission but add a sentence or two to explain that the ‘categorical’ approach leads to the same conclusions than the more complex ‘dummy' approach. I apologise to the authors for raising this technical point as it does not seem to affect their conclusions.

Authors’ response: Thank you very much for your careful attention. We have added a sentence that refers to this matter in the paragraph right after the statistical model is presented. Specifically, we wrote: “Although an ordered structure was considered when designing the intervention packages, we included the intervention package indicators as dummy variables, as the effects of the interventions might not necessarily be ordered.” 

Also, we added following sentence to the discussion section, right before the limitations subsection: “Although the intervention packages were ordered from simple to complex, the observed effects did not necessarily change this way.”

A minor comment to add. Reduce p-values to 2 significant figures in Table 2 (and elsewhere). For example, ‘0.831 to 0.83’ and ‘0.968 to 0.99’. However, for p-values of less than 0.001 the notation p <0.001 is acceptable. 

Authors’ response: It is important to preserve the consistency in using the decimal throughout the document. We have now used 2 decimals for all p-values as well. Thank you!

There appear to be some misprints in the table as a p-value given as ‘0.001>’ should be ‘<0.001’.

Authors’ response: Thank you for catching the errors. We corrected them.

Reviewer #2: The study presents a well-defined and appropriately conducted research problem, objective and methods. 

However, the interventions carried out are not clearly presented (method section). 

Authors’ response: Thank you very much for this helpful comment. We have now replaced Fig. 3 with a new figure and added details and graphical scheme for packages.

The results contain a dense arrangement of data that is also unclear. I suggest that the presentation of the results is simple and summarized. There is a lot of information and it confuses reading.

Authors’ response: We greatly appreciate your comment as clarity is of the most importance for a research document. In an attempt to address it, we have now moved Tables 5 and 6, which are related to linear or non-dichotomous outcomes, to the appendix (s6 file) and siginifcantly reduced their description to a short paragraph at the end of the results secton. The paragraph is: “Similar analyses with non-dichotomous outcome variables for the studied NCDs MetRFs, presented in the S6 file, showed no statistically significant effects of the interventions. Close-to-significant effect of intervention package C was measured only for systolic blood pressure and BMI, confirming the results for dichotomous outcome variables.”

I also believe that the statistical model presented is not the best. Why use a dummy model? The way the results were presented is very confusing. The first reviewer made this consideration, however in a new round the authors did not adjust the manuscript in order to make it simpler.

Authors’ response: We appreciate your concern and detailed attention to the methodology. Referring to this comment, we have now further described this matter in the paragraph right after the statistical model is presented. Specifically, we wrote: “Although an ordered structure was considered when designing the intervention packages, we included the intervention package indicators as dummy variables, as the effects of the interventions might not necessarily be ordered.” Also, we added following sentence to the discussion section, right before the limitations subsection: “Although the intervention packages were ordered from simple to complex, the observed effects did not necessarily change this way.” 

Therefore, I recommend that the article be rejected and the authors prepare a new version for another magazine in order to provide clarity and lightness in the presentation of information.

7. PLOS authors have the option to publish the peer review history of their article (what does this mean?). If published, this will include your full peer review and any attached files.

Do you want your identity to be public for this peer review? For information about this choice, including consent withdrawal, please see our Privacy Policy.

Reviewer #1: No

Reviewer #2: No

---

## [Editor Report · Decision Letter 2]

20 Sep 2024

The effect of results-based motivating system on metabolic risk factors of non-communicable diseases: a field trial study

PONE-D-23-20016R2

Dear Dr. Moradi-Lakeh,

We’re pleased to inform you that your manuscript has been judged scientifically suitable for publication and will be formally accepted for publication once it meets all outstanding technical requirements.

Kind regards,

Thaís São-João

Academic Editor

PLOS ONE
---

## [Editor Report · Acceptance letter]

1 Oct 2024

PONE-D-23-20016R2 

PLOS ONE

Dear Dr. Moradi-Lakeh, 

I'm pleased to inform you that your manuscript has been deemed suitable for publication in PLOS ONE. Congratulations! Your manuscript is now being handed over to our production team.

Kind regards, 

on behalf of

Dr. Thaís São-João 

Academic Editor

PLOS ONE